# *Microbacterium azadirachtae* CNUC13 Enhances Salt Tolerance in Maize by Modulating Osmotic and Oxidative Stress

**DOI:** 10.3390/biology13040244

**Published:** 2024-04-07

**Authors:** Huan Luo, Chaw Su Win, Dong Hoon Lee, Lin He, Jun Myoung Yu

**Affiliations:** 1Department of Applied Biology, Chungnam National University, Daejeon 34134, Republic of Korea; luohuan_0813@163.com (H.L.); chawsu13.yau@gmail.com (C.S.W.); dnwo0817@gmail.com (D.H.L.); helin3022@gmail.com (L.H.); 2College of Resources and Environment, Southwest University, Chongqing 400716, China

**Keywords:** *Microbacterium azadirachtae*, plant growth-promoting rhizobacteria, salt tolerance, maize, root system architecture, climate change, soil salinity stress

## Abstract

**Simple Summary:**

High salinity poses a threat to crop growth and yield. Increasing evidence suggests that environmentally friendly plant growth-promoting rhizobacteria (PGPRs) can mitigate the negative impacts of salt stress by modulating various molecular, biochemical, and physiological processes. In the present study, *Microbacterium azadirachtae* strain CNUC13 was isolated from maize rhizosphere soil. This strain tolerated up to 1000 mM NaCl and 30% PEG 6000 and showed growth-promoting traits like phosphate solubilization and siderophore and indole-acetic acid (IAA) production. The impacts of *M. azadirachtae* strain CNUC13 on maize (*Zea mays* L.) germination, growth, and development with salinity were further examined. The results showed that seed priming with *M. azadirachtae* CNUC13 strain could protect maize from salt stress by modulating plant growth parameters, photosynthetic efficiency, lipid peroxidation, reactive oxygen species, and antioxidant enzyme activities. This is the first study of *M. azadirachtae* on plant growth enhancement and salt stress tolerance in vivo, and the results indicated the vital contribution of *M. azadirachtae* CNUC13 in alleviating the adverse effects of salinity on maize seedlings.

**Abstract:**

Soil salinization is one of the leading threats to global ecosystems, food security, and crop production. Plant growth-promoting rhizobacteria (PGPRs) are potential bioinoculants that offer an alternative eco-friendly agricultural approach to enhance crop productivity from salt-deteriorating lands. The current work presents bacterial strain CNUC13 from maize rhizosphere soil that exerted several PGPR traits and abiotic stress tolerance. The strain tolerated up to 1000 mM NaCl and 30% polyethylene glycol (PEG) 6000 and showed plant growth-promoting (PGP) traits, including the production of indole-3-acetic acid (IAA) and siderophore as well as phosphate solubilization. Phylogenetic analysis revealed that strain CNUC13 was *Microbacterium azadirachtae*. Maize plants exposed to high salinity exhibited osmotic and oxidative stresses, inhibition of seed germination, plant growth, and reduction in photosynthetic pigments. However, maize seedlings inoculated with strain CNUC13 resulted in significantly improved germination rates and seedling growth under the salt-stressed condition. Specifically, compared with the untreated control group, CNUC13-treated seedlings exhibited increased biomass, including fresh weight and root system proliferation. CNUC13 treatment also enhanced photosynthetic pigments (chlorophyll and carotenoids), reduced the accumulation of osmotic (proline) and oxidative (hydrogen peroxide and malondialdehyde) stress indicators, and positively influenced the activities of antioxidant enzymes (catalase, superoxide dismutase, and peroxidase). As a result, CNUC13 treatment alleviated oxidative stress and promoted salt tolerance in maize. Overall, this study demonstrates that *M. azadirachtae* CNUC13 significantly enhances the growth of salt-stressed maize seedlings by improving photosynthetic efficiency, osmotic regulators, oxidative stress resilience, and antioxidant enzyme activity. These findings emphasize the potential of utilizing *M. azadirachtae* CNUC13 as a bioinoculant to enhance salt stress tolerance in maize, providing an environmentally friendly approach to mitigate the negative effects of salinity and promote sustainable agriculture.

## 1. Introduction

Soil salinization has become a major environmental and agricultural issue globally and is likely to significantly intensify with continued climate change [1,2]. According to the Global Map of Salt Affected Soils (GSASmap) database, more than 833 million hectares of subsoil (30–100 cm) and 424 million hectares of topsoil (0–30 cm) currently have above-normal salinity levels [3]. Hassani, Azapagic, and Shokri [2] identified the continents with the largest salt-affected areas to be Asia, Africa, and Australia. They also predicted further expansion in primary soil salinization of dry areas, including regions in the southwest of United States, South America, southern and western Australia, Mexico, and South Africa, between 2071 and 2100 [1]. Unfortunately, approximately 20% of cultivated and 33% of irrigated land in the world is affected by salinity issues [4]. If these conditions persist, it is estimated that up to 50% of agricultural land may lose its ability to support crop production by 2050 [5]. Additionally, based on a global meta-analysis, 17.3% of crop yields were decreased due to salty water irrigation [6], reportedly causing an economic loss of USD 27.3 billion [7,8].

Under salt-stressed conditions, plants experience osmotic and ionic stresses [9]. Osmotic stress begins immediately after salt exposure because excess sodium (Na^+^) and chloride (Cl^−^) ions (NaCl is the most soluble and abundant salt released) cause low external water potential; the resulting dehydration stress reduces stomatal conductance and photosynthetic enzyme function [10]. These physiological alterations lead to the overproduction of reactive oxygen species (ROS), which damage cell membranes and other cellular components via protein oxidation and lipid peroxidation [11]. Hyperosmotic stress and the toxic effects of Na^+^ and Cl^−^ ions in a high-salt environment delay and inhibit seed germination [12], impair root growth and metabolic processes [13], and influence light harvesting and carbon fixation [14]; thus, they decrease crop development and productivity [15,16].

PGPRs are bio-stimulators that actively enhance salt tolerance in host plants, concurrently improving soil biological activity [17,18]. PGPRs enhance stress tolerance in host plants through direct (within the plant and affecting plant metabolism) and indirect (outside the plant) mechanisms [19]. Key mechanisms include alterations in root morphology, the synthesis of growth hormones (auxin, cytokinin, and gibberellin), nitrogen fixation, mineral solubilization (phosphorus and iron), as well as the production of 1-aminocyclopropane-1-carboxylate (ACC) deaminase, siderophores, osmolytes, antioxidants (superoxide dismutase, peroxidase, catalase), volatile organic compounds and extracellular polymeric substances [20,21,22,23]. Moreover, recent studies have shown that the utilization of PGPRs demonstrates the capacity to reprogram global transcriptional patterns and induce epigenetic modifications in host plants to mitigate abiotic stress [24,25,26].

Previous studies have shown that inocula of salt-resistant *Pseudomonas* sp. or *Glutamicibacter* sp. helped alleviate salt stress in *Suaeda fruticosa* by positively influencing the activities of antioxidant enzymes, thus reducing oxidative stress in the cells [17]. Wu et al. [27] reported that the application of *Bacillus halotolerant* KKD1 during wheat cultivation under salt-stressed conditions significantly enhanced plant growth and salt tolerance by inducing a plant stress response that included modulation of phytohormone levels, regulation of lipid peroxidation, accumulation of betaine, and exclusion of Na^+^. Strain KKD1 also positively affected soil nitrogen and phosphorus contents [27]. Inoculation with other PGPR strains, including *Achromobacter*, *Azospirillum*, *Burkholderia*, *Bacillus*, *Enterobacter*, *Pantoea*, *Rhizobium*, *Pseudomonas*, *Paenibacillus*, *Methylobacterium*, and *Variovorax*, enhanced host plant growth in a high-salt environment [28]. The genus *Microbacterium*, isolated from the plant rhizosphere, has been shown to possess several beneficial PGPR traits, including the production of indole acetic acid (IAA), siderophores, and ACC deaminase, as well as the potential to mitigate abiotic stress [29]. However, the plant growth-promoting traits and the induction of salt resistance in crop plants inoculated with *M. azadirachtae* have not yet been extensively investigated. In this study, we isolated *Microbacterium azadirachtae* CNUC13 from the rhizosphere of maize and demonstrated its ability to promote plant growth and induce salt stress tolerance. The plant growth-promoting traits of strain CNUC13 were evaluated with respect to maize seed germination and improved survival under treatment with a gradient of salt concentrations. The specific effects of strain CNUC13 on maize including its vegetative organs, photosynthetic pigments, osmolytes, ROS, and the activities of antioxidant enzymes under salt-stressed conditions were analyzed.

## 2. Materials and Methods

### 2.1. Rhizobacterium Isolation

Bacterial strain CNUC13 was originally isolated from the rhizosphere soil samples of maize (*Zea mays* L.) as described in our previous study [23]. Briefly, maize root samples were gently washed with tap water to remove loosely adhered soil, and 5 g of root tips was moved to 50 mL 0.8% NaCl (*w*/*v*) solution. After vortexing and sonication three times each for 30 s, microorganisms were removed from the root tips. The suspension was serially diluted and spread on separate nutrient agar (NA) plates. After 3 days of incubation at 28 °C, a single bacterial colony was screened by its phenotype. Purified bacterial isolates were stored in 40% glycerol at −80 °C and used for further analyses.

### 2.2. Molecular Identification of Rhizobacteria

The genomic DNA of strain CNUC13 was extracted using the Quick-DNA Fungal/Bacterial Miniprep Kit (Zymo Research, Orange, CA, USA) in accordance with the kit instruction manual. The molecular identification of CNUC13 was conducted via polymerase chain reaction (PCR) amplification and sequencing of the 16S rRNA region using primers 27F (5′-AGAGTTTGATCCTGGCTCAG-3′) and 1492R (5′-TACGGYTACCTTGTTACGACTT-3′) [30]. The resulting sequences were used to search the NCBI GenBank database (https://www.ncbi.nlm.nih.gov/ 13 January 2022) via BLASTn. Multiple alignments were performed using the default settings in the MEGA software (version X) [31]. The neighbor-joining method was utilized in MEGA X, applying the best-fit model of molecular evolution with 1000 bootstrap replicates computed; bootstrap (BS) values above 80% were considered highly supported. Sequences obtained in the present study were submitted to the NCBI GenBank database.

### 2.3. Screening of Bacterial Isolate CNUC13 for Plant Growth-Promoting Traits

To quantify the production of indole-3-acetic acid (IAA), isolates were cultured in Luria–Bertani (LB) broth supplemented with L-tryptophan (100 mg/L) and assessed using the Salkowski reagent, as previously described [32]. IAA production was determined by measuring absorbance at 540 nm using a spectrophotometer and comparing the readings with a standard curve generated from various concentrations of purified IAA (Daejung Chem, Siheung, Republic of Korea).

For phosphate solubilization screening, 10 μL of bacterial isolates were separately inoculated onto Pikovaskya’s agar medium supplemented with tri-calcium phosphate as the mineral P [33]. Formation of a halo zone around bacterial colonies after incubation at 28 °C for 4 days indicated their capacity for phosphate solubilization.

For the evaluation of siderophore production, 10 μL of each bacterial isolate was inoculated onto NA medium overlaid with Chrome Azurol Sulphonate [34]. The presence of a color change observed after 3 days was indicative of siderophore production.

### 2.4. Analysis of Abiotic Stress Tolerance in Strain CNUC13

An overnight culture of strain CNUC13 was adjusted to an OD620 of 1.0 and then re-inoculated at a 1:100 dilution into 20 mL of fresh LB broth containing polyethylene glycol (PEG) 6000 (15 and 30%) or NaCl (0, 400, 600, 800, or 1000 mM). Cultures were grown at 28 °C with rapid shaking (180 rpm) and sampled every 12 h. Bacterial growth was measured using a spectrophotometer at an optical density of 620 nm. LB broth inoculated with sterilized distilled water (SDW) served as a control.

### 2.5. Bacterial Cultures and Inoculation

The *Microbacterium azadirachtae* stain CNUC13 was cultured on an NA plate at 28 °C. After 24 h, a single colony was transferred into nutrient broth (NB) and incubated for 12 h at 28 °C with rapid agitation. The resulting cells were centrifuged (2400× *g*, 5 min) to form a pellet, and then washed three times with SDW. After appropriate dilution, an OD620 of 1.0 was used in all experiments unless specified otherwise.

### 2.6. Plant Materials and Growth Conditions

Maize seeds (cv. Sinhwangok) were surface sterilized with 70% ethanol for 3 min and 5% sodium hypochlorite solution for 3 min, then rinsed 5 times with SDW. The sterilized seeds were immersed in 20 mL of bacterial suspension (OD620 = 0.01), gently agitated for 1 h at room temperature, and dried for 4 h in a laminar flow hood. The bacterial population on the seeds was estimated by collecting 5 dried seeds and preparing serial dilutions in SDW. One hundred microliters of each dilution was spread on NA plates, and the number of colony-forming units (CFU) was calculated after 2 days of incubation. SDW was used as the non-inoculated control.

### 2.7. Germination Rate

The germination rates of maize seeds were evaluated by placing the seeds in a 90 mm Petri plate containing filter paper soaked in NaCl solutions of varying concentrations (0 mM, 30 mM, 50 mM, and 100 mM). Three replicates of 60 seeds were used for each treatment. The plates were kept in a growth chamber (20 °C/16 °C, 16:8 h, light/dark cycle, ca. 50% relative humidity), and the germination rate was recorded on Days 3, 5, and 7. Germination was determined when the radicle length exceeded 0.5 cm [35], and the germination rate (%) was calculated using the following equation: Germination rate (%) = (number of germinated seeds/total number of seeds) × 100.

### 2.8. Growth and Biomass

Artificial soil was mixed with sand (*v*:*v* = 1:1) to facilitate drainage, and the soil mixture was autoclaved twice (121 °C for 1 h, 24 h pause between cycles). Before planting, the soil mixtures were thoroughly wetted with SDW and spiked with 0, 50, 100, 200, and 300 mM NaCl. Maize seeds, either inoculated with bacteria or the non-inoculated control, were sown in plastic tubes (4 cm × 20 cm,) containing 130 g of pre-treated soil and covered with autoclaved vermiculite. To perform the plant biochemical analyses, we induced salt stress by adding 50 mM NaCl into the soil, a condition previously identified as salinity-induced soil stress for the plants [10]. Each treatment was replicated 50 times, and the pots were arranged in a randomized complete block design. All seedlings were grown for 4 weeks in a growth chamber (20 °C/16 °C, 16:8 h, light/dark cycle, ca. 50% relative humidity) and watered every 2 days with 50 mL of SDW. After harvest, plant roots were rinsed with tap water to remove adherent soil mix. The plant growth and biomass were evaluated by measuring the length of the roots and shoots, as well as their respective fresh weight.

To determine the root proliferation, maize seedlings were harvested and thoroughly rinsed with tap water to eliminate soil residues. The above-ground parts were removed, and then the roots were scanned using a flat-bed scanner (Perfection V850 Pro, Epson, Nagano, Japan). Subsequently, the scanned root images were analyzed using the WinRHIZO image analysis system (Regent Instruments, Inc., Quebec City, QC, Canada). This process generated root system data on total root length, root surface area, and the number of lateral roots.

### 2.9. Measurements of Chlorophyll and Carotenoid Contents

The amount of chlorophyll (Chl) present in the leaves was determined spectrophotometrically as previously described [36]. Briefly, 0.1 g of leaf tissue was ground into a powder using liquid nitrogen and then mixed with 1 mL of 80% acetone. Each mixture was vortexed to homogenize the leaf tissues and then centrifuged at 13,000× *g* for 10 min at 4 °C. The OD of each supernatant was measured at 663.6, 646.6, and 470 nm to quantify Chl a, Chl b, and carotenoids. The concentrations of Chl a, Chl b, and total carotenoids were calculated using the following formulae [37]:[Chl a] = 13.71 × OD_663.6_ − 2.85 × OD_646.6_
[Chl b] = 22.39 × OD_646.6_ − 5.42×OD_663.6_
[Total Chl] = [Chl a] + [Chl b]
[Total carotenoids] = [1000 × OD_470_ − 3.27 × (Chl a) − 104 × (Chl b)]/227

### 2.10. Measurement of Osmoregulatory Substances

The proline contents of maize leaves and roots were quantified in accordance with the method of Bates et al. [38] using different concentrations of L-proline (Samchun, Seoul, Republic of Korea) as standards.

The total soluble sugar contents in leaves and roots were determined as previously described [39]. Briefly, leaves and roots were separately ground in liquid nitrogen, and 0.5 g of powder was homogenized in 10 mL of SDW. The samples were vortexed and then boiled for 1 h in a water bath. Chlorophyll was removed by the addition of 0.1 g of activated charcoal; the samples were boiled again for 30 min and centrifuged at 13,000× *g* for 10 min at 25 °C. A 200 µL aliquot of the supernatant was transferred to a new tube, and 1 mL of 0.2% anthrone was added. Each prepared sample was boiled for 30 min at 97 °C and placed in an ice bath to terminate the reaction. The OD at 620 nm was determined and compared with the ODs of standard curves prepared using different concentrations of saccharose.

### 2.11. Measurement of Oxidative Stress Markers and Antioxidant Enzyme Activities

H_2_O_2_ was separately extracted from leaf and root samples using 80% ice-cold acetone and quantified in accordance with the method of Mukherjee and Choudhuri [40]. Activated charcoal was used for chlorophyll removal, and the absorbance at 590 nm was recorded. The total H_2_O_2_ content was calculated from a standard curve constructed with known concentrations of a H_2_O_2_ standard (Sigma, BCCD5257, St. Louis, MO, USA). Lipid peroxidation was estimated based on thiobarbituric acid reactive substances. The malondialdehyde (MDA) contents of maize leaves and roots was estimated through calculation of the amount extracted from 0.5% (*w*/*v*) thiobarbituric acid and 1% (*w*/*v*) trichloroacetic acid agent, as reported by Cui et al. [41]. The enzyme activities of superoxide dismutase (SOD), peroxidase, and catalase were analyzed using the nitrotetrazolium blue chloride reduction [42], guaiacol colorimetric [43], and H_2_O_2_ [44] methods, respectively.

### 2.12. Statistical Analysis

The significance of data was statistically evaluated using one- or two-way analysis of variance (ANOVA) in GraphPad Prism 10.2.1 (GraphPad Software, San Diego, CA, USA). The data represent the mean values with standard error, and a *p*-value < 0.05 was considered significant in each case. All experiments were conducted multiple times, and the sample size for each group is reported in the figure legends.

## 3. Results

### 3.1. Plant Growth-Promoting Characteristics of CNUC13

In a previous study, 139 bacterial strains were isolated from maize rhizosphere, and the 24 potential PGPR strains were confirmed [23]. Among these strains, CNUC13 had the highest level of IAA production over 7 days (26.57 ± 1.17 µg/mL; Figure 1A) and presented substrate solubilization of tricalcium phosphate (Figure 1B) and siderophore production (Figure 1C). These results suggested that strain CNUC13 was able to provide sufficient nutrients for plant growth [18]. Additionally, this strain tolerated up to 30% PEG6000 osmotic stress (Figure 1D) and 1000 mM NaCl salt stress (Figure 1E), indicating that strain CNUC13 could promote plant growth not only under unstressed conditions but also in abiotic stress conditions, including drought and salinity.

### 3.2. Identification of Strain CNUC13

Sequence comparisons with 16S rRNA gene sequences held in the NCBI databank indicated that strain CNUC13 was closely related to genus *Microbacterium*. The sequence was submitted to GenBank under accession number OM279503. The phylogenetic tree constructed using MEGA X software indicated that CNUC13 clustered within the species *Microbacterium azadirachtae* with 97% similarity (Figure 2). These results indicate that the CNUC13 strain belongs to *M. azadirachtae* based on sequence comparison and phylogenetic analysis.

### 3.3. CNUC13 Increases Maize Germination Rates

Seed germination is a critical stage in seedling establishment that determines the success of crop production. Salt stress has been reported to reduce the rate of germination and delay development [45]. To investigate whether strain CNUC13 improves maize seed germination under salt stress, the seeds were coated with bacterial suspension of CNUC13 (ca. 1.5 × 10^7^ CFUs/seed) or with SDW, and the germination rate was examined in the absence or presence of salt stress (0, 30, 50, and 100 mM NaCl; Figure 3A). In the absence of salt stress, there was no significant difference in the maize seed germination rate with or without CNUC13 inoculation (Figure 3B–D). However, salt stress caused significant reductions in the germination rate (*p* < 0.0001) among control seeds at 50 and 100 mM NaCl compared with 0 mM NaCl. However, CNUC13-treated seeds under salt stress had significantly higher germination rates than the matching control of NaCl concentration at 3, 5, and 7 days post inoculation (Figure 3B–D, *p* < 0.0001). These results indicated that CNUC13 treatment significantly promoted seed germination in a high-salt environment.

### 3.4. Maize Growth Is Enhanced by CNUC13 Treatment

To evaluate the physiological impact of CNUC13 on the maize growth under salt-stressed conditions, biomass and growth of CNUC13-inoculated seedlings were compared with non-inoculated seedlings. NaCl treatment caused a deterioration in maize growth, which influenced shoot length, leaf initiation, and leaf expansion (Figure 4A). Compared with unstressed plants, maize height (shoot and leaf length) was significantly reduced in plants exposed to salt stress (Figure 4B). However, among seedlings placed in soil containing 50, 100, 200, and 300 mM NaCl and exposed to strain CNUC13, above-ground lengths were significantly increased by 21.21%, 34.38%, 61.73%, and 128.35% compared with non-inoculated seedlings, respectively (Figure 4B).

A concentration-dependent decrease in the fresh weight (FW) of the above-ground parts of plants in NaCl-treated soil was also observed (Figure 4C), with reductions of 38.12%, 50.43%, 83.06%, and 93.98% in plants exposed to 50, 100, 200, and 300 mM NaCl salt stress compared with 0 mM NaCl, respectively (Figure 4C). In contrast, seedlings treated with strain CNUC13 demonstrated significantly higher above-ground fresh weight compared with the control (Figure 4C); the increases were 30.58%, 66.09%, 88.85%, 137.47%, and 349.22% under 0, 50, 100, 200, and 300 mM NaCl, respectively. Taken together, these results indicate that while salt stress significantly diminished plant height and fresh weight in maize, the application of CNUC13 mitigated the adverse effects of salt stress and increased plant biomass under salt-stressed conditions compared to the non-inoculated control.

### 3.5. CNUC13 Regulates Primary Root Growth and Salt Tolerance in Maize

Plant root systems serve as the interface between plants and soil, and they are used for water acquisition to support plant growth and development [46]. To determine the effect of CNUC13 on maize root development under salt-stressed conditions, root system architectures of seedlings were examined using the WinRHIZO analysis system. As shown in Figure 5A, maize root development and the emergence of lateral roots and root tips were significantly increased by CNUC13 application. In the absence of salt stress, CNUC13 treatment increased the number of root tips by 177.78% compared with the non-inoculated control (Figure 5B), while no significant difference was found in total root length (Figure 5C). In the presence of 50, 100, 200, and 300 mM NaCl salt stress, root length decreased by 55.09%, 78.29%, 94.42%, and 95.95%, respectively, compared with the non-stressed condition (Figure 5C). In contrast, in the presence of 50, 100, 200, and 300 mM NaCl, CNUC13 treatment significantly increased root length by 63.65%, 78.54%, 223.44%, and 149.76%, respectively (Figure 5C) compared with the non-inoculated control seedlings. Similarly, CNUC13 inoculation significantly promoted root tip development among plants grown in 0, 50, 100, 200, and 300 mM NaCl-treated soil; the development increased by 177.77%, 58.47%, 94.64%, 148.13%, and 130.60%, respectively, compared with non-inoculated control seedlings under corresponding salinity stress (Figure 5B). Overall, these results suggest that CNUC13 inoculation of maize significantly promotes root system architecture development, particularly under salt stress, which may contribute to plant growth and salt tolerance by increasing water and nutrient uptake.

### 3.6. CNUC13 Increases Photosynthetic Pigment Production

Salt stress is often associated with photo-inhibition through the reduction in chlorophyll (Chl) and carotenoid contents in the leaves [47]. To investigate whether CNUC13 modulates the biosynthesis of photosynthetic pigments, leaf Chl and carotenoid contents were quantified in the absence or presence of salt stress. In the absence of salt stress, CNUC13 treatment increased Chl a, Chl b, total Chl, and carotenoid contents by 36.32%, 60.51%, 43.07%, and 36.83%, respectively, compared with the levels in control plants (Figure 6A–D). Under 50 mM salt stress, the capacities of maize to produce Chl a, Chl b, total Chl, and carotenoids were significantly reduced by 39.99%, 46.64%, 41.85%, and 42.65%, respectively, compared with the absence of salt stress (Figure 6A–D). However, CNUC13 treatment restored the contents of photosynthetic pigments nearly to levels observed in unstressed controls (Figure 6A–D). These results indicated that CNUC13 increases the photosynthetic capacity of maize in the absence and presence of salt stress, supporting the use of CNUC13 to enhance plant tolerance against salt stress in salinity soils.

### 3.7. CNUC13 Decreases Osmotic Stress in Maize Seedlings

Next, the effects of CNUC13 on the contents of osmoregulatory substances, proline, and soluble sugar in the leaf and root were analyzed. In the absence of salt stress, there were no significant differences in proline contents between control and CNUC13-treated plants (Figure 7A). In response to salt stress (50 mM NaCl), proline contents of control plants were significantly increased by 465.12% in leaves and 122.28% in roots compared with the non-stressed plants (Figure 7A). However, proline contents were markedly less increased by 33.2% in CNUC13-inoculated seedlings than control plants under salt-stressed conditions (Figure 7A). The soluble sugar content was significantly reduced by 48.23% and 41.69% in salt-stressed leaves and roots, respectively. However, in CNUC13- inoculated plant leaves, the soluble sugar content was significantly increased by 38.97% and 56.5% in the absence and presence of salt stress, respectively (Figure 7B). These results suggest that treatment with CNUC13 can improve the osmo-protective properties of maize cells, protecting them from salt stress-related damage.

### 3.8. CNUC13 Protects against Oxidative Stress in Maize Seedlings

Elevated levels of Na^+^ and Cl^−^ are detrimental to multiple cellular processes because they induce reactive oxygen species (ROS) production in the cell and alter protein–protein interactions [9,48,49]. To determine whether CNUC13 inoculation modulates ROS accumulation and oxidative stress in maize seedlings, the levels of hydrogen peroxide (H_2_O_2_) and malondialdehyde (MDA) were evaluated. In control seedlings, salt stress significantly increased H_2_O_2_ production in leaves by 40.47% compared with plants grown in unstressed soil (Figure 7C). However, in the presence of salt stress, H_2_O_2_ production in leaves was significantly reduced by CNUC13 treatment compared with the control plants. The levels of MDA, an oxidative stress indicator, were also significantly increased by salt stress, both in the leaves (77.57%) and roots (39.2%), compared with non-salt-stressed plants (Figure 7D). However, in the presence of salt stress, CNUC13 treatment induced a significant decrease by 24.49% in leaf MDA levels compared with the control. These results suggest that CNUC13 treatment reduces salt stress-induced toxic ROS production and oxidative stress, thereby increasing salt tolerance in maize.

### 3.9. CNUC13 Normalizes the Activities of Antioxidant Enzymes in Maize Seedlings

Considering that oxidative stress damages the plant cellular environment [48,50], the effect of CNUC13 on antioxidant enzymes among plants grown in the presence or absence of salt stress was investigated. As shown in Figure 8A–C, salt stress induced significant increases in the enzymatic activities of catalases (19.45% and 36.39%), superoxide dismutases (26.45% and 5.07%), and peroxidases (37.96%) in the leaves and roots of maize. However, CNUC13 treatment significantly decreased the levels of ROS-scavenger enzymes in the presence of 50 mM NaCl, such that they became similar to the levels in the unstressed controls (Figure 8A–C). These results indicate that CNUC13 protects plants from salt stress-induced osmotic and oxidative stresses.

## 4. Discussion

High salinity in agricultural soils significantly reduces crop growth and productivity globally, impacting various crops including broccoli, cauliflower, common bean, cowpea, and wheat [51,52,53,54]. Plants experiencing salinity stress undergo diverse morphological and physiological changes, resulting in reduced growth and development, as well as disruptions in reproductive processes such as germination and vegetative growth [28]. Moreover, soil salinity negatively affects plant biochemical processes, triggering oxidative damage through the generation of reactive oxygen species (ROS), promoting ion toxicity, nutrient reduction, modulating antioxidant processes and phytohormone imbalances, and decreasing photosynthetic rates [28,55,56]. Besides the various salt stress-alleviating applications, the utilization of PGPR has emerged as a promising and environmental-friendly method for enhancing salt stress tolerance in plants.

The present study evaluated the effects of the newly isolated PGPR strain *M. azadirachtae* CNUC13 on the growth and development of maize plants under salt stress. High salinity caused osmotic stress (Figure 7A,B) and oxidative damage (Figure 7C,D), thus inhibiting maize seed germination (Figure 3), decreasing photosynthetic pigment production in maize seedlings (Figure 6), and reducing plant growth (Figure 4). In contrast, inoculation with strain CNUC13 limited the negative effects of salt stress, significantly improving maize germination and growth rates. Strain CNUC13 was also capable of producing high levels of IAA and siderophore as well as tricalcium phosphate solubilization (Figure 1), which may play important roles in geochemical carbon pools, nitrogen cycles, and nutrient availability. Increased levels of IAA may also contribute to the regulation of root development and root length (Figure 5), allowing for maize to maintain sufficient nutrient uptake from the soil.

PGPRs enhance salt tolerance in plants through direct growth-promoting effects [20], such as rendering nutrients accessible to plants, solubilizing phosphorus and potassium, producing siderophores to facilitate iron uptake, and fixing atmospheric nitrogen [57,58]. Additionally, PGPRs produce various plant growth hormones, such as IAA, cytokinin, gibberellins, and ethylene, which promote plant growth [59]. Similarly, the increases in root biomass induced by IAA-producing *Azospirillum* spp. enhance plant nutrient uptake, thereby stimulating bacterial colonization and amplifying the inoculation effect [60]. Elevated levels of IAA also govern the initiation of lateral roots and expansion of the root surface area, allowing for plants to effectively absorb nutrients from the soil [18]. However, Xie et al. [61] found that although root elongation was related to bacterial IAA, additional mechanisms were involved in PGPR-mediated growth promotion. Based on PGP trait screening results, CNUC13 can produce siderophores in addition to IAA production. The presence of salinity in the soil reduces the solubility of trace elements, thereby leading to a scarcity of accessible iron for plant uptake [62]. However, PGPR-producing siderophores, such as iron chelating agents, can benefit plants by aiding in iron acquisition. Iron chelation is involved in regulating plant metabolic processes, including chlorophyll synthesis, the electron transfer system, and nucleic acid synthesis [62,63]. Therefore, a balanced interplay of different PGPR traits may be needed to boost plant growth [64]. Effective IAA production combined with siderophore production and phosphorus solubilization would explain the enhanced growth as well as improved salt tolerance observed in CNUC13-treated maize.

Under salt stress, plants generate and accumulate osmotic regulators (osmolytes), including proline and sugars, to maintain intracellular osmotic pressure homeostasis [28,65]. Proline is considered as a salt stress indicator in plants due to its role as an ROS scavenger. It also functions to regulate the redox potential and stabilize proteins, membrane structures, and the electron transport system involved in photosynthesis in plants [56]. In our study, we observed a significant increase (*p* < 0.001) in the proline contents in maize seedlings under salt-stressed conditions. Conversely, a decrease in proline levels was noted with CNUC13 inoculation (Figure 7A). Similarly, previous studies have demonstrated an increase in proline content due to salinity stress, while inoculation with PGPR has been shown to reduce proline levels in plants [66,67,68,69]. This suggests that a decrease in proline accumulation indicates successful mitigation of salt stress through the application of PGPR. Additionally, other studies have also shown that proline deposition in the root tips of maize under a low water potential is the result of increased proline transport from the leaves to roots rather than from direct proline biosynthesis in the roots [70,71]. Similarly, our results showed significantly greater proline accumulation in the leaves than in the roots under salt stress (Figure 7A), suggesting that salinity-induced proline production is primarily synthesized in the leaves and then transported to the roots to mitigate the abiotic stress. CNUC13 inoculation significantly reduced the proline content in maize subjected to salt stress, consistent with previous findings [65,72,73] that bacterial inoculation significantly decreased proline accumulation in salt-stressed plants. Thus, plants inoculated with PGPR may be partially protected from high salt stress, leading to reduced proline accumulation [65,72,73].

Similar to proline, soluble sugars are another key modulator in plant osmoregulation [9,49,74], and their levels vary according to NaCl concentration. Wang et al. [75] reported that the sugar level in the leaves of maize seedlings increased with 150 mM NaCl but significantly decreased with 300 mM NaCl at 5 days post seeding. Our results showed that 50 mM NaCl decreased the accumulation of soluble sugars in maize leaves and roots at 4 weeks after planting (Figure 7B). Shoot growth reduction during salt stress occurs in two phases: a rapid response to the increase in external osmotic pressure (“osmotic stress phase”) and a slower response caused by the accumulation of Na^+^ in the leaves (“ion toxicity phase”) [9]. Accordingly, prolonged exposure (4 weeks) of maize plants to 50 mM NaCl, and thus to high ion (Na^+^ and Cl^−^) concentrations, presumably induced the second phase of salt stress (ion toxicity) to hinder normal regulatory functions, including soluble sugar synthesis. Conversely, CNUC13 application increased soluble sugar accumulation in maize under salt stress. Previous studies have also revealed that PGPR increased the soluble sugar content during abiotic stress and improved plant resistance to salinity-induced osmotic stress [56].

Increased cellular ROS production is a common response to abiotic stresses, causing severe oxidative damage to plants including degradation of chlorophyll, protein degeneration, DNA damage, and lipid peroxidation of cellular membranes [28,76]. MDA is a final metabolite produced by ROS-mediated cellular damage; therefore, the level of MDA content is considered a biomarker for the ROS-mediated cellular membrane breakage [77]. Our results showed that in plants inoculated with CNUC13 under salt-stressed conditions, the concentrations of H_2_O_2_ and MDA were significantly decreased compared to those in control plants (Figure 7C,D). Similarly, *Bacillus firmus* SW5 inoculated soybean plants showed reduced levels of H_2_O_2_ and MDA under salt-stressed conditions [78]. Oxidative stress in plants under environmental stress is mitigated by antioxidant defense systems [48], demonstrated in the leaves of maize plants in this study, where salt stress increased the levels of catalase, peroxidase, and SOD enzymes (Figure 8). Catalase and peroxidase reduce H_2_O_2_ to H_2_O and O_2_ [79,80]; their levels, together with the level of SOD, increase in response to salt stress-induced ROS production. However, CNUC13 treatment decreased, rather than increased, the levels of these three enzymes in maize leaves exposed to 50 mM NaCl. A possible explanation for this finding is that higher antioxidant enzyme activities at earlier time points removed excessive ROS, resulting in lower enzyme levels at the studied time point.

## 5. Conclusions

This study demonstrated that *Microbacterium azadirachtae* strain CNUC13, isolated from maize rhizosphere soil, exhibits multiple PGP traits including phosphate solubilization and the production of IAA and siderophores. Inoculation of CNUC13 on maize seed enhanced germination rates, as well as the growth and development of maize seedlings under salt stress. The protective effect of CNUC13 could be attributed to the integration of multiple physio-biochemical processes, such as enhanced photosynthetic pigment synthesis (chlorophyll and carotenoids), reduced accumulation of osmotic (proline) and oxidative (ROS and MDA) stresses, and normalized antioxidant enzymatic activities (catalase, SOD, and peroxidase). Our results indicate that CNUC13 inoculation can serve as a sustainable and eco-friendly strategy to improve agricultural production in high-salinity soils. This study also provides insights concerning the induction of salt tolerance by *M. azadirachtae* strain CNUC13, which has not been previously studied as a PGPR strain.

## Figures and Tables

**Figure 1 biology-13-00244-f001:**
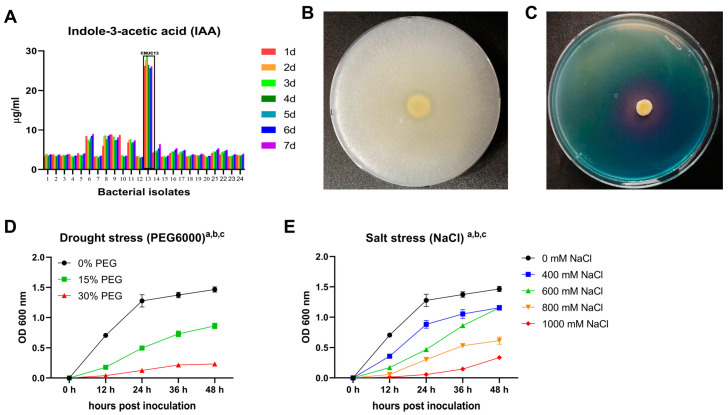
Plant growth-promoting traits and abiotic stress tolerance of *Microbacterium azadirachtae* CNUC13. (**A**–**C**) CNUC13 produced the highest level of IAA among other rhizobacteria over 7 days (**A**) and was capable of tricalcium phosphate solubilization (**B**) and siderophores production (**C**). (**D**,**E**) CNUC13 tolerated up to 30% PEG6000 osmotic stress (**D**) and 1000 mM NaCl salt stress (**E**). Data in (**D**,**E**) are expressed as the mean  ±  standard error of the mean (SEM, *n*= 4–5). Tukey’s multiple comparisons analyses: a, main effect of incubation time (*p* < 0.05); b, main effect of abiotic stress (*p* < 0.05); c, main effect of interaction (*p* < 0.05).

**Figure 2 biology-13-00244-f002:**
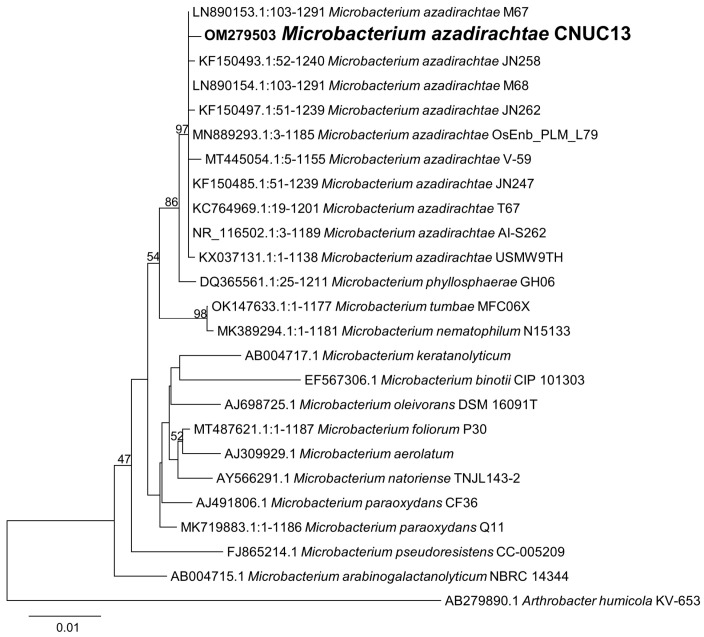
Phylogenetic analysis of *Microbacterium azadirachtae*. Evolutionary relationships of *M. azadirachtae* CNUC13 were inferred using the neighbor-joining method, with *Arthrobacter humicola* KV-653 as the outgroup. Evolutionary distances were calculated as the nucleotide substitutions per site using the Kimura 2-parameter method.

**Figure 3 biology-13-00244-f003:**
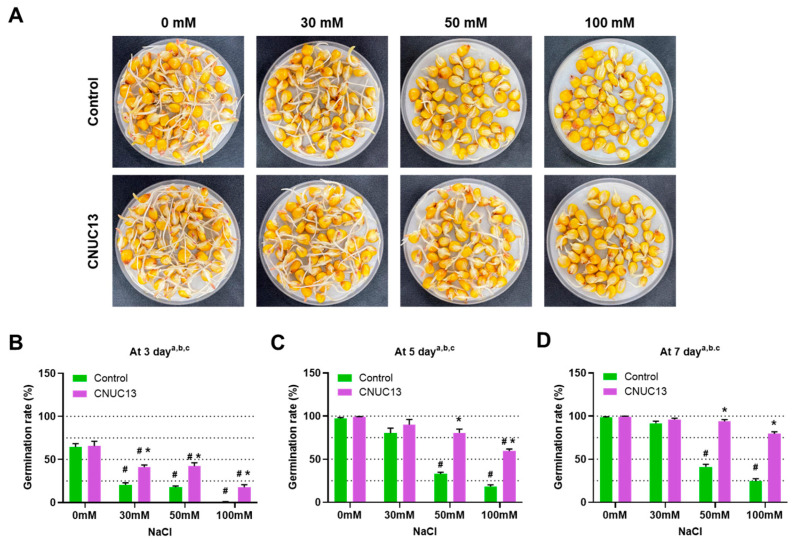
*Microbacterium azadirachtae* CNUC13 enhances maize germination under salt stress. (**A**) Representative images of maize seed germination 7 days post sowing on Petri dishes containing 0, 30, 50, and 100 mM NaCl. (**B**,**D**) Maize seeds (60 seeds per treatment) were coated with SDW or a suspension of strain CNUC13; the germination rate was measured in the absence or presence of salt stress at 3 (**B**), 5 (**C**), or 7 (**D**) days. Data are expressed as the mean  ±  SEM. Tukey’s multiple comparisons analyses; a, main effect of salt stress (NaCl, *p* < 0.05,); b, main effect of treatment (CNUC13, *p* < 0.05); c, main effect of interaction (*p* < 0.05). #, based on comparison with 0 mM NaCl in the matching treatment (*p* < 0.01); *, based on the matching control according to NaCl concentration (*p* < 0.05).

**Figure 4 biology-13-00244-f004:**
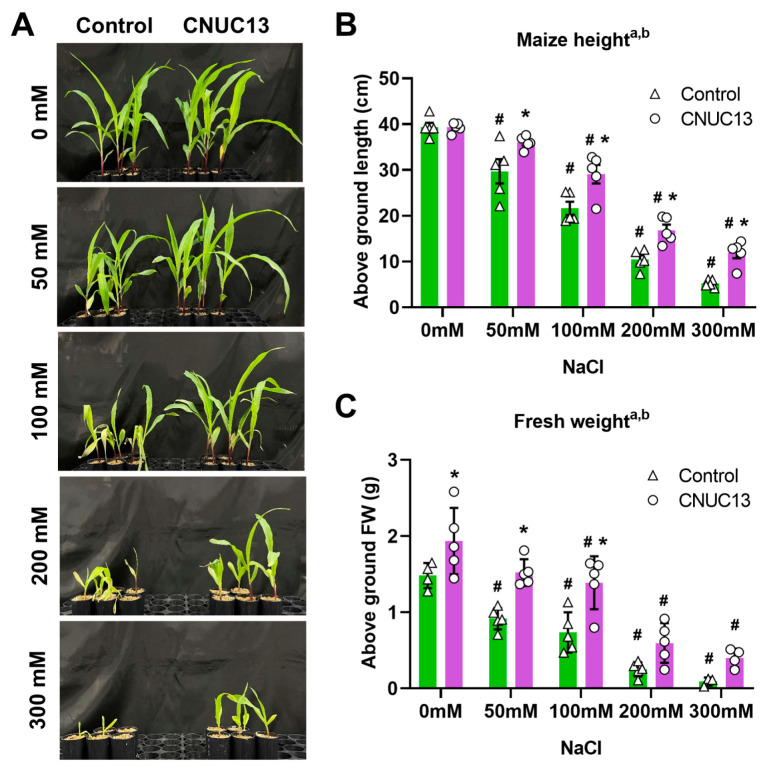
*Microbacterium azadirachtae* CNUC13 enhances maize growth under salt stress. (**A**) Representative images of maize growth after 28 days of seedlings exposure to unstressed or salt-stressed conditions. (**B**,**C**) Whole leaf and shoot length (**B**) and fresh weight (FW (**C**)) were recorded. Data are expressed as the mean  ±  SEM (*n* = 5). Tukey’s multiple comparison analyses: a, main effect of salt stress (NaCl, *p* < 0.05); b, main effect of treatment (CNUC13, *p* < 0.05); #, comparison with 0 mM NaCl in the matching treatment (*p* < 0.01); *, based on the matching control according to NaCl concentration (*p* < 0.01). The experiments were repeated three times.

**Figure 5 biology-13-00244-f005:**
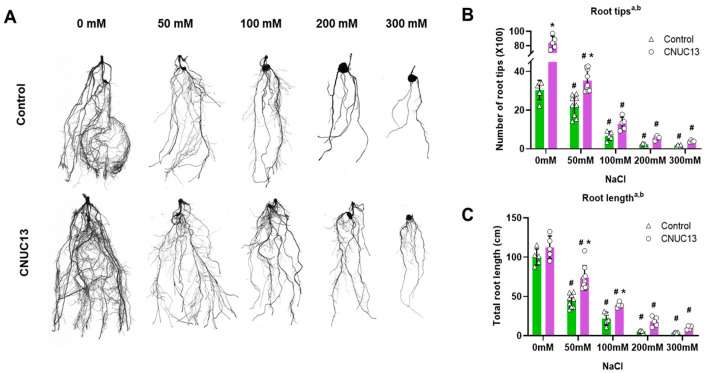
*Microbacterium azadirachtae* CNUC13 protects maize root development under salt stress. (**A**) Representative images of maize root and the emergence of the lateral root were analyzed at 28 days post sowing in the absence or presence of salt stress. Images were analyzed using the WinRHIZO system. (**B**,**C**) Salt stress induced a significant decrease in the number of root tips (**B**) and total root length (**C**), whereas CNUC13-treated roots showed significant increases in length and tip number compared with the control. Data are expressed as the mean  ±  SEM (*n* = 4–9). Tukey’s multiple comparison analyses: a, main effect of salt stress (NaCl, *p* < 0.05); b, main effect of treatment (CNUC13, *p* < 0.05). *, based on the matching control according to NaCl concentration (*p* < 0.05); #, based on comparison with 0 mM NaCl in the matching treatment (*p* < 0.01). The experiments were repeated three times.

**Figure 6 biology-13-00244-f006:**
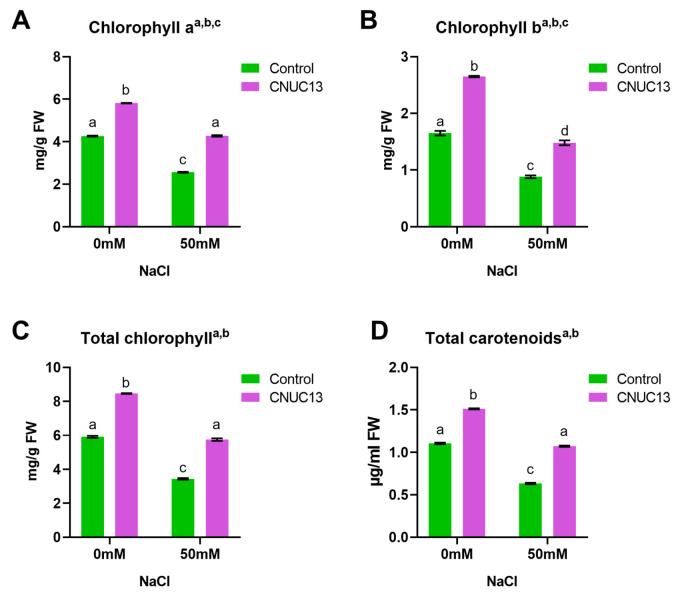
Salt stress-induced photo-inhibition is alleviated by CNUC13 treatment. The contents of leaf chlorophyll (**A**–**C**) and carotenoids (**D**) were measured in maize grown in normal or NaCl-treated soil. NaCl induced significant decreases in leaf chlorophyll and carotenoid contents, whereas CNUC13 treatment increased the photosynthetic capacity in maize. Data are expressed as the mean  ±  SEM (*n* = 5). Tukey’s multiple comparison analyses: a, main effect of salt stress (*p* < 0.05); b, main effect of CNUC13 treatment (*p* < 0.05); c, main effect of interaction (*p* < 0.05). Different letters on columns indicate significant differences between groups. The experiments were repeated three times.

**Figure 7 biology-13-00244-f007:**
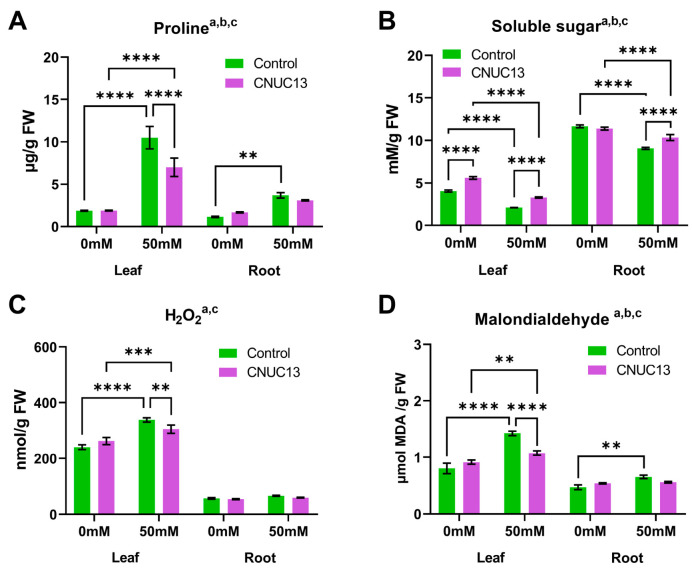
CNUC13 reduces salt stress-induced osmotic and oxidative stresses. Salt stress increases osmotic stress (**A**,**B**) and ROS production (**C**,**D**), whereas CNUC13 treatment reverses both phenomena. Data are expressed as the mean  ±  SEM (*n* = 5). Tukey’s multiple comparisons test: a, main effect of salt stress (*p* < 0.05); b, main effect of CNUC13 treatment (*p* < 0.05); c, main effect of interaction (*p* < 0.05); ** *p* < 0.01; *** *p* < 0.001; **** *p* < 0.0001. The experiments were repeated three times.

**Figure 8 biology-13-00244-f008:**
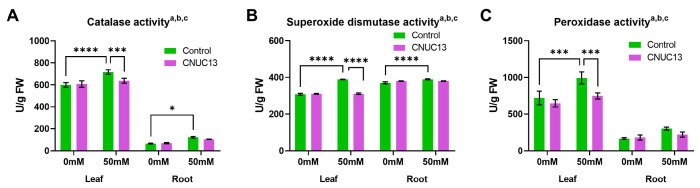
CNUC13 modulates the activities of antioxidant enzymes. Salt stress increased the enzymatic activities of antioxidants, catalase (**A**), superoxide dismutase (**B**), and peroxidase (**C**) while CNUC13 decreased NaCl-induced increases. Data are expressed as the mean  ±  SEM (*n* = 3). Tukey’s multiple comparison analyses: a, main effect of salt stress (*p* < 0.05); b, main effect of CNUC13 treatment (*p* < 0.05); c, main effect of interaction (*p* < 0.05); * *p* < 0.05; *** *p* < 0.001; **** *p* < 0.0001. The experiments were repeated three times.

## Data Availability

The datasets used and/or analyzed during the current study are available from the corresponding author on reasonable request.

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
