# Peer review of "Microbacterium azadirachtae CNUC13 Enhances Salt Tolerance in Maize by Modulating Osmotic and Oxidative Stress"

_biology, 2024, doi:10.3390/biology13040244_

Round 1

Reviewer 1 Report

Comments and Suggestions for Authors

Title: Microbacterium azadirachtae CNUC13 Enhances Salt Tolerance in Maize by Modulating Osmotic and Oxidative Stress

This study presents an investigation into the potential of the plant growth-promoting rhizobacterium (PGPR), Microbacterium azadirachtae strain CNUC13, to alleviate salt stress in maize. Soil salinization poses a significant threat to global food security, and the use of PGPRs represents an eco-friendly strategy to enhance crop productivity in affected areas. The authors isolated strain CNUC13 from maize rhizosphere soil, demonstrating its tolerance to high concentrations of NaCl and PEG 6000, alongside PGPR traits such as indole-3-acetic acid (IAA) production, siderophore and phosphate solubilization. Through phylogenetic analysis, the strain was identified as M. azadirachtae. The study further explored the effects of inoculating maize plants under salt stress with CNUC13, noting improvements in germination rates, seedling growth, biomass, root architecture, photosynthetic pigments, and a reduction in stress indicators. The treatment was also found to enhance antioxidant enzyme activities, suggesting a mechanism by which CNUC13 confers salt tolerance to maize.

Overall, this manuscript makes a contribution to the field of agricultural biotechnology by demonstrating the potential of M. azadirachtae CNUC13 to enhance salt tolerance in maize. The findings are robust, significant, and pave the way for further research into PGPRs as a sustainable solution to soil salinization challenges. With some revisions, particularly in deepening the discussion on the mechanisms of action and practical application considerations, this study could have a substantial impact on the scientific community and agricultural practices.

The manuscript would benefit from a more detailed discussion comparing CNUC13's efficacy with other known PGPRs in similar contexts.

The details regarding the reproducibility of the experiments, such as the number of replicates and statistical analysis methods, could be elaborated to strengthen the study's credibility.

A discussion on the feasibility of large-scale application, including considerations for inoculant production, application methods, and cost-effectiveness, would greatly enhance the manuscript's impact.

More recent studies that highlight the ongoing challenges and advancements in the field of PGPR research and its role in combating soil salinization should be added in the introduction section

Some figures and tables could be optimized for clarity and comprehensiveness.

There are typographical and grammatical errors throughout the manuscript that need correction. Additionally, ensuring consistency in terminology and units of measurement will improve the manuscript's professionalism and readability.

Comments on the Quality of English Language

There are typographical and grammatical errors throughout the manuscript that need correction. Additionally, ensuring consistency in terminology and units of measurement will improve the manuscript's professionalism and readability.

Author Response

We greatly appreciate the thorough review and constructive comments provided by the editor and reviewers. All comments have been carefully addressed, and necessary changes to the text have been implemented as recommended. Please find the attached file. 

Reviewer 2 Report

Comments and Suggestions for Authors

Author Response

We greatly appreciate the thorough review and constructive comments provided by the editor and reviewers. All comments have been carefully addressed, and necessary changes to the text have been implemented as recommended.

Reviewer 3 Report

Comments and Suggestions for Authors

1.   Line 14: “Zea mays L.”.

2.   In the abstract, significant numeric data should be presented, e.g. by how many percents the traits were improved?, etc. Moreover, the methodology of the current study is not well presented. Furthermore, the abstract also contains a very wordy sentence (32-37) which should be shortened or separated into shorter ones. Although the meaning of study has been presented in the Simple Summary, it should be also discussed in the end of the abstract along with some potential further applications based on the current study.

3.   The concentration of the bacteria used in the experiment should appear in the abstract as well.

4.   The Internet citation should appear in the references and numbered as the other citations (lines 50-51)

5.   The manuscript contains too many run-on sentences which make the it wordy and cause reduced readability. For example, lines 56-50, 69-74, 77-81, etc.

6.   Lines 56-60: this should be cited.

7.   In the second paragraph of the introduction, there should be more literature review about the damages of salinity to  agricultural production.

8.   The Microbacterium genus should be introduced more to lead to the novelty statement of the study, then there should be hypotheses.

9.   Some of the maize cultivar’s characteristics should be presented.

10.                       In section 2.4, the method for evaluation of drought should add more details

11.                       In the results, there should not be citations. Lines 261-264, 320-321,355-356; 415; method should delete in line 290-292

12.                       “Two-way ANOVA, Tukey's multiple comparisons test” should not be in the results.

13.                       The first paragraph of the discussion is not necessary. The arguments in this paragraph should be placed at the beginning of the other paragraphs in the discussion.

14.                       The conclusion should indicate the significant numbers of the study. The conclusion should give the main findings.

Comments on the Quality of English Language

Minor editing of English language required

Author Response

(The authors gave the same response as above.)

Round 2

Reviewer 2 Report

Comments and Suggestions for Authors

Comments to Authors’ response for Manuscript ID biology-2925038. 

The article entitled as “Microbacterium azadirachtae CNUC13 Enhances Salt Tolerance in Maize by Modulating Osmotic and Oxidative Stress”. 

The authors provided comprehensive answers to the questions. 

However, there is a remark that does not call into question the significance of the work. The note concerns the response to comment No. 5. Even with the naked eye it is clear that bacteria significantly improve seed germinated. However, in order not to mislead readers, an explanation of the photograph Figure 3A-100mm Control should be given in the text: to indicate the principle by which it was assessed that the seeds had sprouted. Below is this photo. Figure 3A are photographs of germinated seeds after 7 days of seed germination. For 100mm Control, germinated seeds are completely invisible (about 0%). However, in the Figure 3D-100mm Control graph (shown below), which presents the numerical values of the same results presented in the photo, germination is about 75%.
